# Structural basis for DNA unwinding at forked dsDNA by two coordinating Pif1 helicases

Nannan Su [1,2,6], Alicia K. Byrd [3,6], Sakshibeedu R. Bharath[2], Olivia Yang[4], Yu Jia[1,2], Xuhua Tang[2], Taekjip Ha [4]*, Kevin D. Raney[3]* & Haiwei Song [1,2,5]*

Pif1 plays multiple roles in maintaining genome stability and preferentially unwinds forked dsDNA, but the mechanism by which Pif1 unwinds forked dsDNA remains elusive. Here we report the structure of *Bacteroides sp* Pif1 (BaPif1) in complex with a symmetrical double forked dsDNA. Two interacting BaPif1 molecules are bound to each fork of the partially unwound dsDNA, and interact with the 5′ arm and 3′ ss/dsDNA respectively. Each of the two BaPif1 molecules is an active helicase and their interaction may regulate their helicase activities. The binding of BaPif1 to the 5′ arm causes a sharp bend in the 5′ ss/dsDNA junction, consequently breaking the first base-pair. BaPif1 bound to the 3′ ss/dsDNA junction impacts duplex unwinding by stabilizing the unpaired first base-pair and engaging the second base-pair poised for breaking. Our results provide an unprecedented insight into how two BaPif1 coordinate with each other to unwind the forked dsDNA.

[1] Life Sciences Institute, Zhejiang University, 388 Yuhangtang Road, Hangzhou 310058, China. [2] Institute of Molecular and Cell Biology, 61 Biopolis Drive, Singapore 138673, Singapore. [3] Department of Biochemistry and Molecular Biology, University of Arkansas for Medical Sciences, Little Rock, AR 72205, USA. [4] Department of Biophysics and Biophysical Chemistry, Johns Hopkins University, 725N. Wolfe Street, Baltimore, MD 21205, USA. [5] Department of Biochemistry, National University of Singapore, 14 Science Drive, Singapore 117543, Singapore. [6]These authors contributed equally: Nannan Su, Alicia K. Byrd *email: tjha@jhu.edu; raneykevind@uams.edu; haiwei@imcb.a-star.edu.sg

Helicases are modular motor proteins that couple the binding and hydrolysis of ATP to unwinding of double-stranded (ds) DNA and play essential roles in nearly all aspects of nucleic acid metabolism, ranging from DNA replication to chromatin remodeling[1,2]. Helicases are classified into six different superfamilies based on the conserved helicase motifs, which are further divided into two subgroups, A and B according to the directionality of their translocation on single-stranded (ss) DNA[1].

*Saccharomyces cerevisiae* Pif1 (ScPif1), the founding member of the Pif1 family helicases[3], is a superfamily 1B (SF1B) helicase that has multiple functions in the maintenance of genomic homeostasis[4,5]. ScPif1 is a negative regulator of de novo telomere formation and elongation by strongly inhibiting telomere addition to the ends of both chromosomes and double-strand breaks (DSBs)[6–9]. ScPif1 contributes to DNA replication in many ways, which include processing Okazaki fragments[10–13], promoting fork progression at hard-to-replicate regions such as highly transcribed tRNA genes[14,15] and G-quadruplex (G4) structures[16–18], and stimulating repair-associated DNA synthesis with Polδ during break-induced replication[19,20].

Pif1 is evolutionarily conserved from bacteria to humans[21]. Pif1-like DNA helicases including ScPif1 and Rrm3, fission yeast Pfh1, human Pif1 (hPif1) and BaPif1 contain a degenerate sequence located between motifs II and III termed the Pif1 signature motif[22]. The signature motif of ScPif1 is essential in vivo for mitochondrial and nuclear functions and in vitro for ATPase activity[23] while the signature motif of Pfh1 is necessary for both protein displacement and helicase unwinding activities[24]. It has been suggested that the Pif1 signature motif exerts its functional role indirectly through stabilizing the residues involved in ssDNA binding[25] but the underlying mechanism remains elusive.

The in vitro activities of Pif1 proteins have been biochemically characterized extensively. ScPif1 requires a single-stranded DNA (ssDNA) region for binding and robustly unwinds forked dsDNA, RNA/DNA hybrids (R-loops) and some G4 structures but is very non-processive when unwinding conventional 5′-tailed dsDNA and very stable G4 structures[3,17,26–28]. Pif1 unwinds duplex DNA in a single-base-pair kinetic step, powered by the hydrolysis of one ATP molecule[29].

While Pif1 is a monomer in solution, it can form a stable dimer on DNA[30]. Dimerization is not required for the helicase to translocate efficiently on ssDNA[31] or to unwind substrates under conditions where annealing is limited[32]. For unwinding forked dsDNA substrates, ScPif1's helicase activity can be stimulated by the non-translocating strand at the 3′-end[3,32]. These suggest the possibility that multiple Pif1 molecules bind to the forked dsDNA for efficient unwinding, but how these Pif1 molecules coordinate with each other for unwinding the forked DNA is unclear.

We and others have solved several crystal structures of Pif1 including BaPif1[25,33], ScPif1[34], and human Pif1[25,35]. All these structures with/without ATP analog and/or ssDNA have improved our understanding of the helicase activity of Pif1. However, in the absence of dsDNA bound structures of Pif1 helicases, the effect of specific interactions with the duplex on the unwinding mechanism remains poorly understood.

To reveal the structural basis for unwinding dsDNA by Pif1, we determined the structure of BaPif1 in complex with a symmetrical dual forked dsDNA and ADP·AlF$_4^-$. The structure showed that two Pif1 molecules bind to the 5′ arm and 3′ ss/dsDNA junction, respectively. The structural data combined with mutagenesis together with the kinetics and single-molecule FRET (smFRET) analysis provide key mechanistic insights into forked dsDNA unwinding by BaPif1.

## Results

**Structural overview.** Since ScPif1 can form a stable dimer on a forked dsDNA[30], we reasoned that a symmetrical dual forked dsDNA would allow multiple Pif1 molecules to bind such a substrate, thereby facilitating crystallization. We crystallized BaPif1 in complex with ADP·AlF$_4^-$ and a symmetrical dual forked dsDNA containing a 10 nt oligo-dT at both 3′ and 5′ ends (designated as BaPif1–10dT-fdsDNA; Fig. 1a) and determined its structure at a resolution of 3.3 Å. The asymmetric unit of the crystal structure contains two BaPif1 molecules plus one DNA strand, thus the model of the biologically meaningful BaPif1–10dT-fdsDNA complex is generated by the application of a crystallographic two-fold operation. The structure of BaPif1–10dT-fdsDNA showed that two BaPif1 molecules bind to each fork of 10dT-dsDNA and interact with their respective forks in the same manner (Fig. 1b). For simplicity, we only describe the BaPif1–DNA interactions at one fork, which involves molecules A and B of BaPif1. Out of the 10 base-pairs in the double-stranded region of DNA, only eight well-ordered base-pairs could be modeled (Supplementary Fig. 1a). A total of two base-pairs, one on each side of this central double-stranded region are broken. In the single-stranded regions, 7 dT could be modeled at the 5′ end while only 4 dT at the 3′ end were observed. The conformations of two BaPif1 molecules are essentially identical and are similar to those observed in our previous structures of BaPif1 in complex with ssDNA and ADP·AlF$_4^-$ [25], suggesting that the BaPif1 molecules in these structures undergo similar conformational change upon concomitant binding of ATP and DNA (Supplementary Fig. 1b). The two BaPif1 molecules bound to the junction regions of one fork are related by a rotation of 68° rather than 180°. Therefore, the two BaPif1 molecules bound to the junction regions of a fork cannot be considered as a conventional dimer.

**Interaction of BaPif1 with DNA.** As shown in Fig. 2, the binding of BaPif1 to the 5′ arm causes a sharp bend between T10 and C11; consequently, the base of T10 points in the direction opposite to those of other oligo-dTs. The flipped base of T10 is sandwiched between Tyr91$^A$ and Tyr412$^B$. It should be noted that the Tyr91$^A$ belongs to the BaPif1 bound to the 5′ arm while Tyr412$^B$ is contributed by the BaPif1 bound to the 3′ ss/dsDNA junction. A similarly flipped base was observed in the structure of BaPif1-dH but it stacks against only Tyr91[25]. Importantly, the bending in the 5′ ss/dsDNA junction breaks the first C11–G20 base-pair. The base of unpaired C11 is sandwiched between Ile409$^B$ from BaPif1 bound to the 3′ arm and the base of adjacent G12. In the 3′ ss/dsDNA junction; the unpaired G20 is stabilized by residues Val149$^B$ and His361$^B$, which stack against the base and ribose ring of G20, respectively. Another notable feature is that three residues His236$^B$, His377$^B$ and Phe379$^B$ from BaPif1 bound to the 3′ ss/dsDNA are juxtaposed at the helical junction of ss/dsDNA junction. Phe379$^B$ stacks against the base of C19 in the second G12–C19 base-pair while His236$^B$ and His377$^B$ contact the ribose rings of C19 and G12, respectively. Moreover, Lys151$^B$ and Lys152$^B$ from BaPif1 bound to the 3′arm interact with the phosphate backbone of the double-helical region of DNA, therefore anchoring BaPif1 in close proximity to the 3′ ss/dsDNA junction. Among residues involved in 3′ ss/dsDNA recognition, only V149$^B$, Lys151$^B$, His361$^B$, and Phe379$^B$ are conserved across Pif1 family helicases (Supplementary Fig. 2), suggesting that they might be critical for unwinding.

**Interaction of two BaPif1 molecules at ss/dsDNA junctions.** At the ss/dsDNA junctions, the interaction of two BaPif1 molecules buries a total solvent accessible surface area of 716 Å$^2$. BaPif1

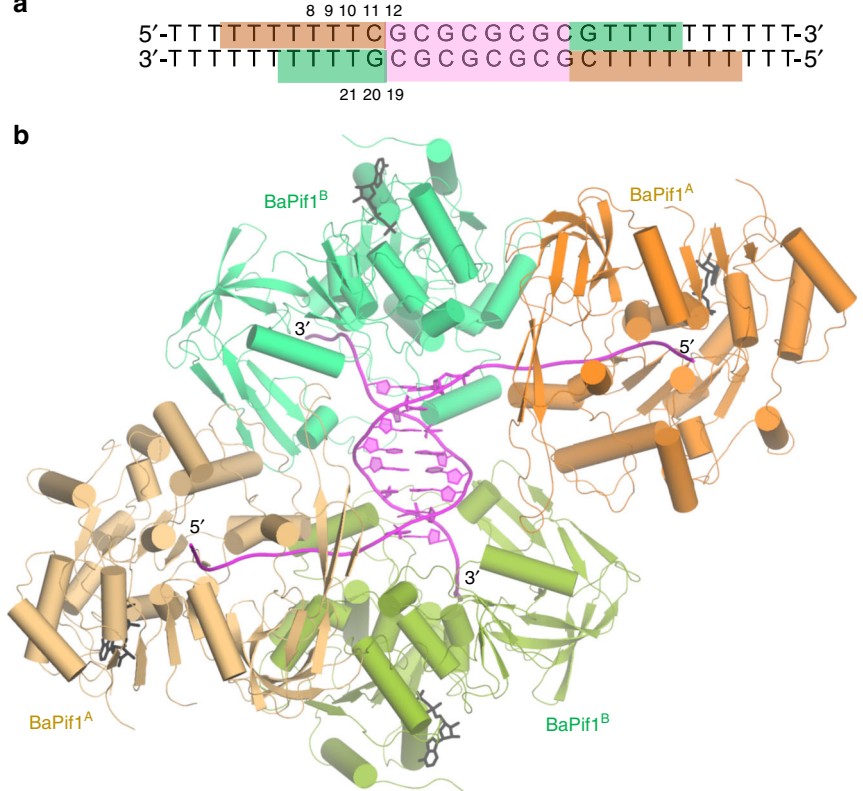

**Fig. 1** Structure of BaPif1 complexed with a dual forked dsDNA. **a** DNA sequence used for crystallization: The eight out of the ten nucleotide base pairs of the central dsDNA observed in the structure are shown in violet. The brown and green blocks correspond to nucleotides bound to BaPif1 at 5′ and 3′ arms of the dual forked dsDNA. The nucleotides discussed in the text are numbered. **b** Structure of BaPif1 bound to dual forked dsDNA: Molecules at 5′ arms labeled as BaPif1[A] and shown in two shades of brown, are related by a two-fold symmetry axis. Similarly, molecules at 3′ arms labeled as BaPif1[B] and shown in two shades of green, are related by the same two-fold axis. The asymmetric unit consists of two molecules of BaPif1 (one in green and the other in brown) and one strand of DNA. ADP·AlF$_4^-$ used in crystallization is shown as black sticks.

bound to the 5′ arm uses loops β11–β12, β12–β13 and β14 of the 2B domain to interact with loop α8–α9, helices α9 and α13 of the 2A domain in the BaPif1 bound to the 3′ ss/dsDNA junction predominantly through a network of salt bridge contacts and hydrogen bonds (Fig. 3). Key interactions include salt bridges between Lys321[A] and Glu210[B], Glu323[A] and Lys405[B], and hydrogen bonds between Asn308[A] and Asp209[B], Leu291[A] and Gln205[B], Glu323[A] and Ser208[B]. These interactions are further strengthened by Van der Waal's contacts involving residues Lys292[A], Val309[A], Asp209[B], Pro410[B] and Tyr412[B]. Interestingly, the residues involved in BaPif1[A]–BaPif1[B] interaction are not well conserved among Pif1 helicases (Supplementary Fig. 2). The weak nature of the interface may allow two BaPif1 molecules bound to the 5′ and 3′ arms translocate in the opposite directions easier, therefore coordinating with each other and regulating their activities.

**Mutational analysis of residues in the ss/dsDNA junction.** The structure of BaPif1–dT10-fdsDNA showed that two BaPif1 molecules bind to the ss/dsDNA junctions. As Pif1 is a 5′–3′ helicase and has critical functions in DNA replication, we designed a substrate to determine whether the BaPif1 molecules bound to both arms of the fork are both active helicases, capable of unwinding duplex DNA. This substrate mimics a stalled DNA replication fork, which contains dual duplexes termed parent and leading duplexes (Fig. 4a). This DNA substrate enables us to examine the unwinding of dual duplexes simultaneously, hence the coordination of the activities of the two BaPif1 molecules can be discerned. Wild type BaPif1 unwinds both the forked (parent)

duplex and the non-forked (leading) duplex on the 3′-arm of the fork although the amplitudes of product formation are low even at saturating enzyme concentrations (Supplementary Fig. 3). Interestingly, more products are formed from the leading duplex than the parent duplex.

The greater product formation observed with leading duplex may be due to it being only 12 bp whereas the parent duplex contained 16 bp. A substrate was tested which has 16 bp duplexes on each arm of the substrate (Fig. 4b). The quantity of product for the dual duplex with 16 bp was similar to that for the substrate containing the 12 bp leading duplex (Fig. 4c–e) which indicates that the leading duplex is preferentially unwound.

To examine the role of residues contacting the ss/dsDNA junction, we made three single Ala mutants (H236A, H377A, and F379A) and one double mutant (Y91A/Y412A). Neither H236A nor F379A showed unwinding activity for either of the duplexes in a single turnover reaction while both H377A and Y91A/Y412A variants exhibited markedly reduced unwinding activities (Fig. 4c, e). We also measured the binding of H236A and F379A to ssDNA (Supplementary Fig. 4). Both variants still bind DNA so the mutations have not affected protein folding or prevented them from binding to the substrates. However, the affinities are moderately reduced. Hydrolysis of ATP on ssDNA is markedly reduced for the H236A, F379A, and Y91A/Y412A variants (Supplementary Fig. 4), which could explain the low unwinding activity. In addition to interacting with the ss/dsDNA junction, residues His236, His377 and Phe379 in the BaPif1 bound to the 5′ arm interact with 5′ ssDNA (Supplementary Fig. 5) while they stabilize the broken first base-pair and are poised to assist

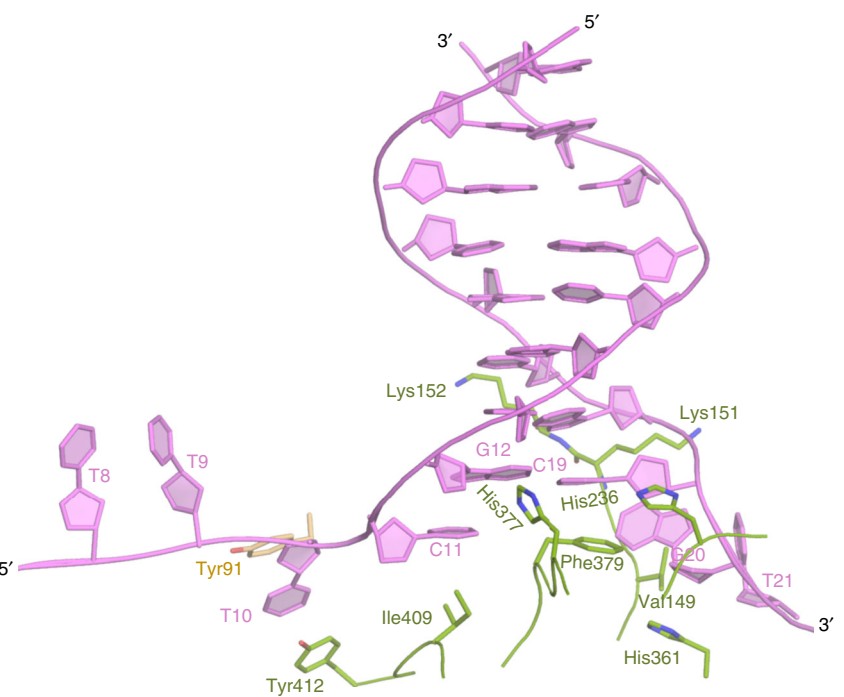

**Fig. 2** Interaction of BaPif1 with forked dsDNA. Residue in brown belongs to BaPif1[A], residues in green belong to BaPif1[B] and nucleotides of the forked dsDNA are in violet. Tyr91[A] and Tyr412[B] stack against T10. The first base pair (C11–G20) of the dsDNA is broken and C11 is stacked on one side by Ile409[B] and on another side by the subsequent nucleotide G12. The other nucleotide, G20 of the broken base pair is stabilized by Val149[B] and His361[B]. The main chain of Lys151[B]–Lys152[B] anchor the phosphate backbone of the 3′ arm. A trio of residues, Phe379[B], His236[B] and His377[B] interact with the G12–C19 base pair of the dsDNA.

breaking the second base-pair in the BaPif1 bound to the 3′ ss/dsDNA junction (Fig. 2). Mutations of these three residues not only inhibit unwinding of the leading duplex but also affect the unwinding of the parental duplex.

Interestingly, BaPif1[B] bound on the 3′-ssDNA interacts with the junction similarly to the SF1A DNA helicases PcrA and UvrD that translocate in the opposite direction on ssDNA compared to BaPif1. The stacking interaction of Phe379 against the second base-pair in BaPif1 bound to 3′ ss/dsDNA junction is reminiscent of those observed for Phe626 and Tyr621 in SF1A DNA helicases PcrA[36] and UvrD[37], respectively (Supplementary Fig. 6). Phe626 of PcrA and Tyr621 of UvrD are located in a Pin region (termed as a wedge for dsDNA unwinding) and stack against the first base-pair in the ss/dsDNA junctions. Structural data combined with mutational analysis suggested that these two residues directly participate in duplex unwinding driven by ATP-coupled 3′-5′ translocation. In contrast, although Phe379 in BaPif1 bound to the 3′ arm also stacks with a base-pair at the ss/dsDNA junction, it may exert its role in unwinding indirectly through assisting the unwinding activity of the BaPif1 bound to the 5′ arm.

The flipped base on the 5′ arm is sandwiched between Tyr91 of BaPif1 bound to the 5′ arm and Tyr412 of BaPif1 bound to the 3′ ss/dsDNA junction (Figs. 2, 3). Together they stabilize the bent ssDNA at 5′ ss/dsDNA junction. As the ssDNA bending caused by BaPif1 bound to the 5′ arm is a key event for breaking the first base-pair, mutation of these two Tyr residues to Ala would be expected to destabilize the bent conformation of substrate DNA, thereby impacting both the ATPase and unwinding activities of BaPif1. Indeed, the Y91A/Y412A variant exhibits reduced rates of ATP hydrolysis (Supplementary Fig. 4). The unwinding activity of the Y91A/Y412A variant is also greatly reduced (Fig. 4c, e), which is not surprising as unwinding is driven by ATP hydrolysis-dependent translocation in a 5′-3′ direction.

**Mutational analysis of the BaPif1–BaPif1 interface**. Mutations of residues in the BaPif1[A]–BaPif1[B] interface would not be predicted to directly affect the enzymatic activities of BaPif1. Indeed, some of these Pif1 variants have only small changes in activity relative to the wild type enzyme (D209A and D209A/K405A) (Fig. 4d, e). However, the E323A/K405A variant unwinds more DNA in a single turnover reaction than the wild type enzyme for both the parental and the leading duplex and eliminates the preference for unwinding the leading duplex (Fig. 4d, e). The quantities of product formed are equivalent for the two duplexes for E323A/K405A. This may indicate that the two BaPif1 molecules regulate each other and in the absence of interaction between the two BaPif1 molecules, each is able to unwind and is only limited by the inherent processivity of the enzyme.

We also measured the unwinding activity of the E323K variant using the dual duplex substrate (Fig. 4d, e). The E323K variant showed increased product formation in a single turnover reaction relative to the wild type. The activity of the E323K variant is similar to that of the E323A/K405A. These data suggest that the increase in activity observed in the E323A/K405A variant is due to the mutation of E323 more so than due to the mutation of K405. This is also consistent with the reduction in activity of D209A/K405A relative to D209A, which also suggests that a K405A mutation does not increase activity. Gel filtration analysis of wtBaPif1 or E323K complexed with a single forked DNA at a molar ratio of 2:1 (protein:DNA) in the presence of ADP·AlF$_4^-$ showed that less 2:1 and more 1:1 BaPif1–DNA complexes were formed for E323K than wtBaPif1 (fractions 22–25), indicating that dimerization on DNA is weakened for the E323K variant (Supplementary Fig. 7). In support of this notion, E323[A] forms a salt bridge with K405[B] and hydrogen bonds with S208[B] (Fig. 3). The rates of ATP hydrolysis by both the E323A/K405A and E323K variants are only slightly increased relative to the wild type enzyme (Supplementary Fig. 4), The small increase in ATP

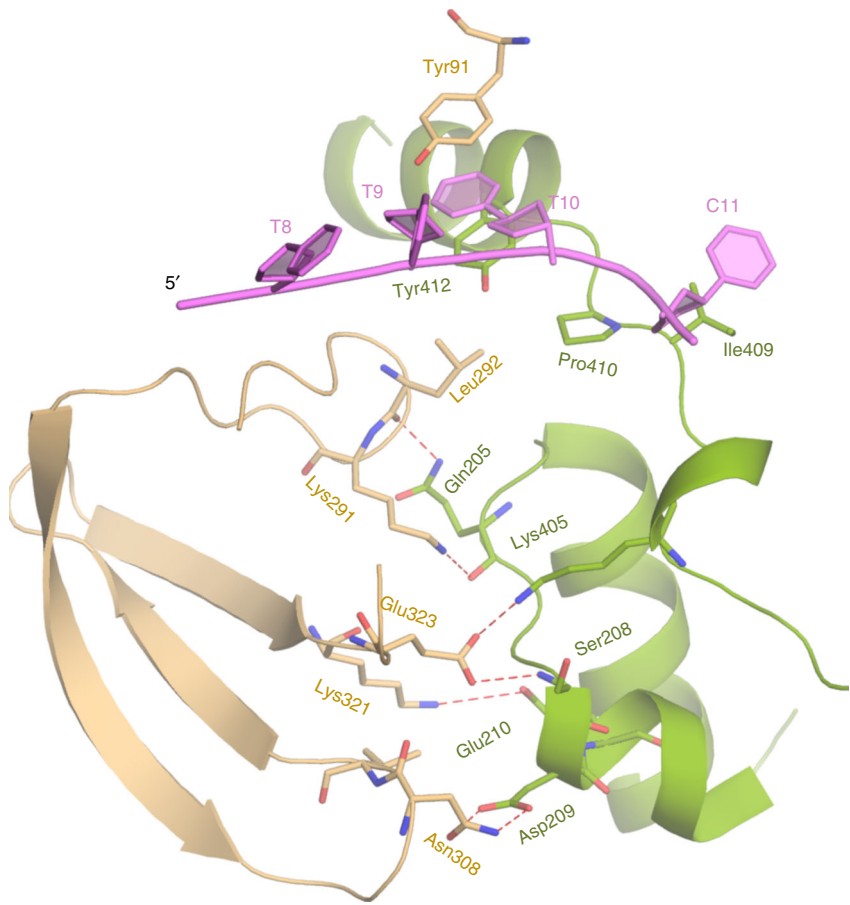

**Fig. 3** Interaction between BaPif1[A] and BaPif1[B] of the unwinding complex. Salt bridges between Glu323[A]-Lys405[B] and Lys321[A]-Glu210[B] are further strengthened by hydrogen bonds between Asn308[A] and Asp209[B], Glu323[A] and Ser208[B], Leu291[A] and Gln205[B], and Van der Waal's interactions involving Lys292[A], Val309[A], Asp209[B], Pro410[B], and Tyr412[B].

hydrolysis is not sufficient to explain the 2-fold increase in unwinding of the leading duplex and 3-fold increase in unwinding of the parental duplex by these variants.

**Mutational analysis using smFRET.** To confirm the kinetic observations for Y91A/Y412A and E323A/K405A, single-molecule FRET (smFRET) microscopy and alternating laser excitation (ALEX) were used to simultaneously track the loss of Cy3 labeled leading duplex and Cy5 labeled parental duplex (Fig. 5a). Upon addition of BaPif1 or its variants with ATP to the substrate, the traces (Fig. 5b and Supplementary Fig. 8) allowed us to estimate the fraction of population for initially unwinding either parental or leading duplexes (Fig. 5c), and the fraction of duplexes unwound by wtBaPif1, Y91A/Y412A and E323A/K405A (Fig. 5d), and also the rate of unwinding leading or parental duplexes (Fig. 5e). Y91 from BaPif1[A] and Y412 from BaPif1[B] both contact the flipped base on the 5′ overhang of the parental duplex so it is not surprising that Y91A/Y412A had a stronger preference for unwinding the leading duplex (Fig. 5c) and the unwinding rate for the parental duplex was much slower than that of wtBaPif1 (Fig. 5e). Consistent with the single turnover kinetics data, E323A/K405A lacked any preference for unwinding either of the duplexes (Fig. 5e) and its apparent unwinding rate was faster than that of wtBaPif1, and both duplexes were unwound at a similar, faster rate (Fig. 5e). We also observed a mid-FRET intermediate state that occurred before unwinding of both parental and leading duplexes (Fig. 5b and Supplementary Fig. 8). This intermediate was present in 27% of unwound molecules for both wtBaPif1 and E323A/K405A mutants, but

more frequent (60%) for Y91A/Y412A. The lifetime of E323A/ K405A intermediates were much shorter whereas the lifetime of Y91A/Y412A intermediates was about double that of wtBaPif1 (Fig. 5f). The slower unwinding rate of the parental duplex and longer lifetime of intermediates for Y91A/Y412A strongly support the notion that these two residues are important for stabilizing the bent ssDNA at 5′ ss/dsDNA junction.

**Two BaPif1 molecules bound to the fork are dually active.** Wild type BaPif1 unwinds the parental and leading duplexes with similar rate constants (Fig. 6). Mutation of E323 and K405 increases both the unwinding rate constants and the quantity of product formed on the parental and leading duplexes (Fig. 6). Importantly, the combined amplitudes for the fraction of unwound product equals 1.2 for this BaPif1 variant. Thus, each molecule of helicase unwinds a duplex simultaneously. The increase in activity upon uncoupling the two BaPif1 subunits suggests that the interaction of two BaPif1 molecules may serve a regulatory role whereby each of the BaPif1 subunits clears the DNA on one strand of the fork but the linkage of the enzymes constrains their activity so that only a small region of the DNA is cleared. This is consistent with the known role of Pif1 in DSB repair[7,38], regulation of telomerase[8], and Okazaki fragment processing[11,12].

Our structure of BaPif1 in complex with 10dT-fdsDNA and unwinding assays showed that two BaPif1 molecules bind to and are active on both arms of the forked dsDNA. Since ScPif1, human Pif1 and BaPif1 are structurally similar and all are capable of unwinding the forked DNA substrates, it is envisaged that two

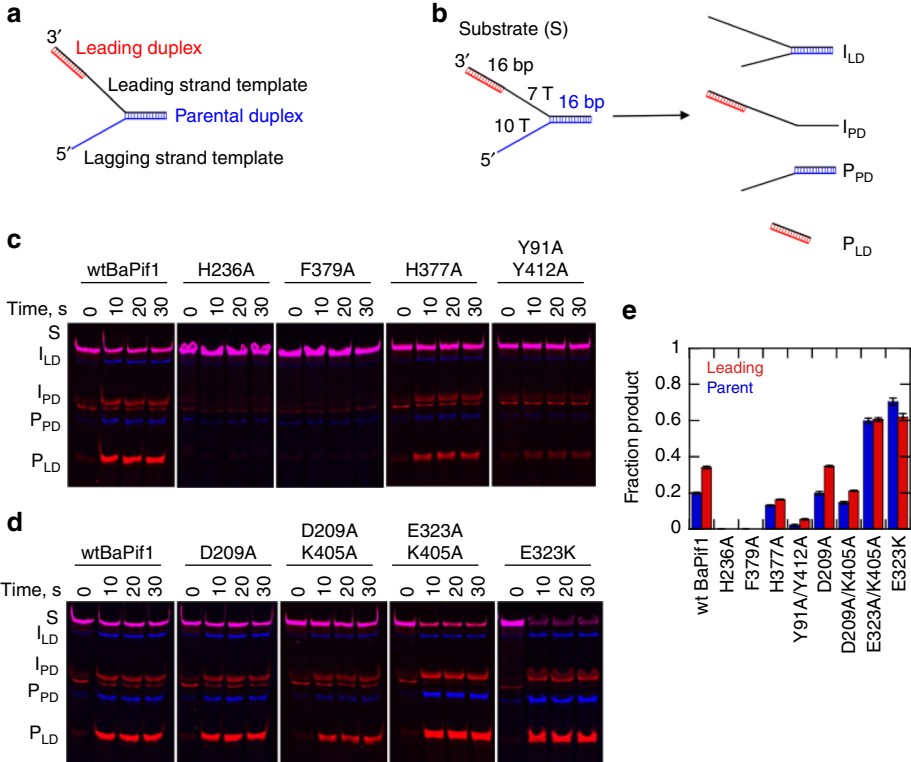

**Fig. 4** Single turnover unwinding of a dual duplex substrate by BaPif1 and its variants. **a** The substrate contains a fluorescein labeled forked 16-bp duplex (blue) called parental duplex and a Cy5 labeled 16-bp duplex on the 3′ arm of the fork (red) termed the leading duplex. **b** The substrate (S), intermediates ($I_{LD}$ with the leading duplex unwound and $I_{PD}$ with the parental duplex unwound) and products ($P_{PD}$ is the trapped product of parental duplex unwinding and $P_{LD}$ is the trapped product of leading duplex unwinding) as separated by native PAGE. **c** Unwinding assay with wild type and mutants, H236A, F379A, H377A, and Y91A/Y412A. **d** Unwinding assay with wild type and interface mutants, D209A, E323A/K405A, D209A/K405A and E323K. **e** Quantification of the products formed from leading/parental duplex region in single turnover unwinding assays with wild type BaPif1 and its variants. Values are the average product formation and standard deviation of three independent experiments. Source gels and quantification data are provided as a Source Data file.

molecules of both ScPif1 and hPif1 would bind and unwind the same substrate we used for BaPif1. Consistent with this notion, unwinding assays (Supplementary Fig. 9) showed that under single turnover conditions, ScPif1 unwinds both duplexes but with strong preference for the forked parental duplex (Supplementary Fig. 9a–c). Addition of a short tail to the displaced strand of the leading duplex allowed ScPif1 to unwind both duplexes similarly to BaPif1 (Supplementary Fig. 9d–f), consistent with ScPif1's preference for unwinding forked duplexes[3,29,32]. hPif1 produced no product with either substrate in a single turnover reaction; however, multi-turnover experiments demonstrated that hPif1 unwinds both duplexes (Supplementary Fig. 9g–i). The different unwinding efficiencies and preferences for substrate duplexes for ScPif1, hPif1, and BaPif1 may be attributed to the different duplexes to be unwound by them in their respective species. Further structural evidence of ScPif1 and hPif1 in complex with forked dsDNA are required to prove this view.

**Mechanism of forked dsDNA unwinding by BaPif1.** For unwinding the 5′ tailed dsDNA, previously we proposed based on the structure of BaPif1-dH that the bending at the 5′ ss/dsDNA junction caused by the simultaneous binding of ADP·AlF$_4^-$ and a 5′ tailed dsDNA would break the first few base pairs and the rest of a short dsDNA region is likely to be unwound by thermal fraying[25]. In support of this model, our current structure shows that the binding of BaPif1 to the 5′ arm causes a similar bending and consequently breaks the first base-pair. Based on our structural observations and available biochemical data, we propose a

mechanism for unwinding a forked dsDNA by BaPif1 helicase (Fig. 7). Two BaPif1 molecules synergistically bind to both arms of the fork. The BaPif1 molecule bound to the 5′ arm causes the breaking of the first base-pair through bending the tracking strand. The second BaPif1 molecule bound to the 3′ arm interacts extensively with the ss/dsDNA junction such that it not only stabilizes the first broken base-pair but also is poised to assist breaking the second base-pair. The binding of BaPif1 to the 3′ arm may also prevent re-winding the separated strands. Further unwinding of the duplex is carried out by ATPase coupled translocation of the molecule at the 5′ arm. Given that the BaPif1 bound to the 3′ ss/dsDNA junction can only translocate in a 5′–3′ direction, the model we propose herein suggests that two BaPif1 molecules bound to the fork, initiate unwinding, after which each BaPif1 monomer can function as an active helicase. However, the protein dimer ensures only short stretches of duplex are unwound, keeping with many of the reported functions of Pif1 family enzymes. Consistent with this model, biochemical studies showed that the 3′ ssDNA tail stabilizes a dimer formed on a forked dsDNA[30] and stimulates the unwinding activity but hinders the annealing activity of ScPif1[3,32,39]. Moreover, our unwinding assays showed that the two BaPif1 helicases bound to the fork are dually active (Figs. 4, 6 and Supplementary Fig. 3). Interestingly, reduction of the interactions between the monomers results in increased unwinding processivity suggesting that the protein-protein interactions may serve a regulatory mechanism whereby each BaPif1 molecule influences the activity of the other. When the protein-protein interactions are reduced, the

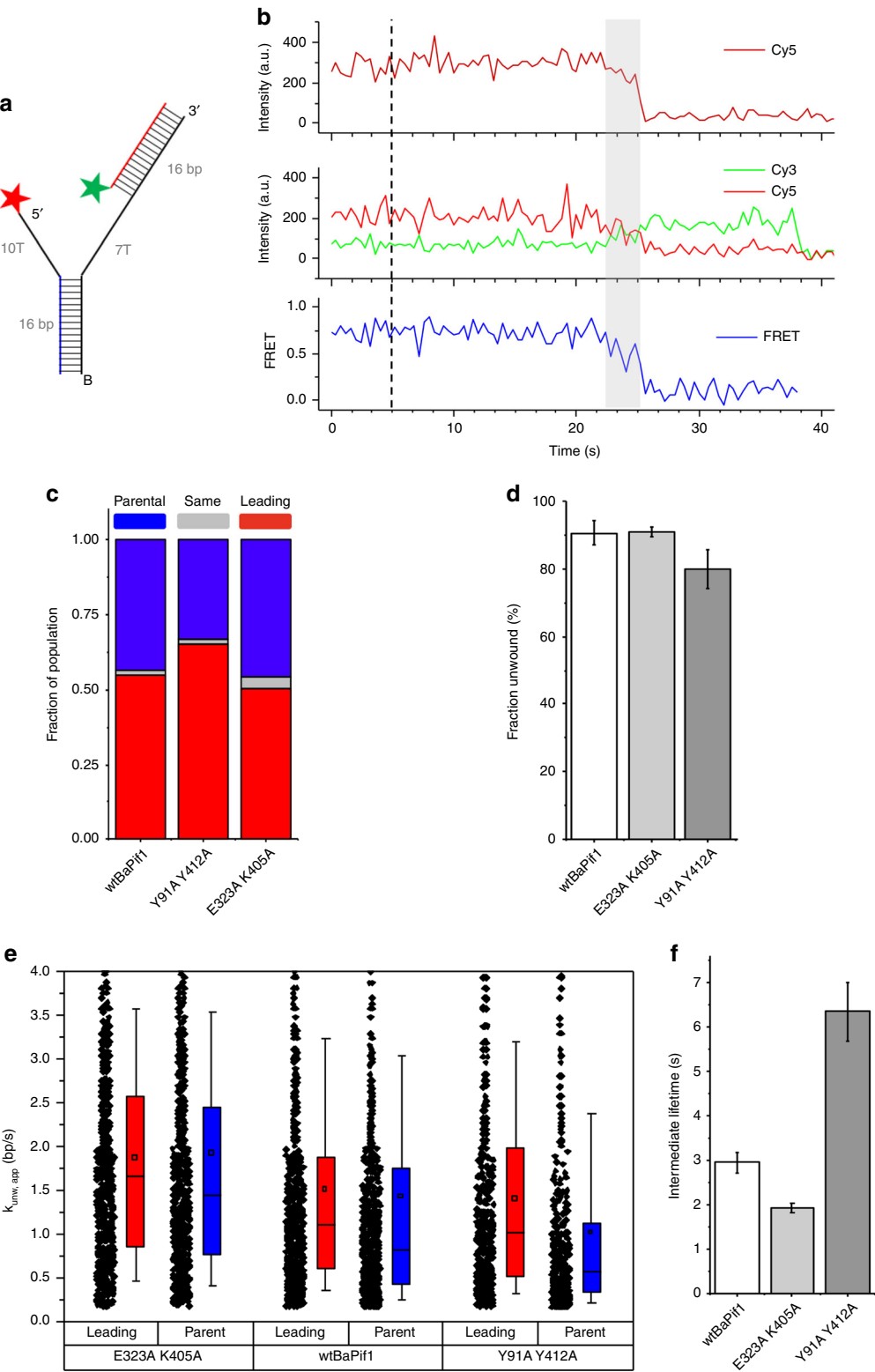

ability of each monomer to translocate forward unhindered is increased, which could result in more processive unwinding by the E323 variants.

To date, the structures of only monomeric forms of SF1 and SF2 helicases in complex with a dsDNA have been reported[36,37] although a dimeric UvrD has been proposed to be required for processive unwinding of dsDNA based on single turnover kinetic data[40]. In the proposed dimeric UvrD–DNA complex, one

monomer binds to the 3′ ss/dsDNA junction while the second monomer is bound to the 3′ ssDNA tail, and two interacting UvrD molecules form a stable initiation complex. Moreover, UvrD can self-assemble to form dimers and tetramers in the absence of DNA whereas ScPif1 without bound DNA is in a monomeric form in solution[30]. However, a dimeric UvrD with or without bound DNA has not been crystallized. Therefore, our structure of BaPif1–dT10-fdsDNA represents the only one

**Fig. 5** smFRET experiments showing unwinding of a dual duplex substrate by BaPif1 and its variants. **a** The substrate is labeled with Cy5 (red star) on the 5′ end of the fork to observe unwinding of the parental duplex (blue) and with Cy3 (green star) on the 3′ end of the leading duplex (red) to observe leading duplex unwinding. The substrate was anchored to the single-molecule surface with a biotinylated (black B) DNA. **b** Example smFRET (bottom) trace of unwinding of parental duplex before leading duplex. Mid-FRET intermediate state (gray) observed with Cy3 excitation (middle). Unwinding of parental duplex (loss of signal) observed with direct Cy5 excitation (top), shortly after appearance of intermediate (gray). BaPif1 and ATP were flown in at 5 s (dashed line). **c** Fraction of unwound substrates where parental duplex is unwound first (blue), or leading duplex is unwound first (red), or both are unwound simultaneously (gray) for wtBaPif1, Y91A/Y412A, and E323A/K405A. **d** Fraction of substrates unwound calculated as $100 - \langle \frac{n_{initial}}{n_{final}} \rangle$, where $n_{initial}$ is the number of complete unwinding substrates (Fig. 5a) per imaging area before addition of BaPif1 and ATP, and $n_{final}$ is the number of complete unwinding substrates per imaging area after addition of BaPif1 and ATP, and averaged over $n = 3$ repeat experiments. Errors bars are standard deviation. **e** Apparent unwinding rate of parent (blue) and leading (red) duplexes for wtBaPif1, E323A/K405A, and Y91A/Y412A. Mean (square marker), median (horizontal line), 25th–75th percentile (box), and 10th–90th percentile (whiskers) have been indicated. Data points are also shown. **f** Lifetime of mid-FRET intermediate states of wtBaPif1 and variants E323A/K405A, and Y91A/Y412A (based on **b** and Supplementary Fig. 7 gray areas). Errors bars are standard deviations.

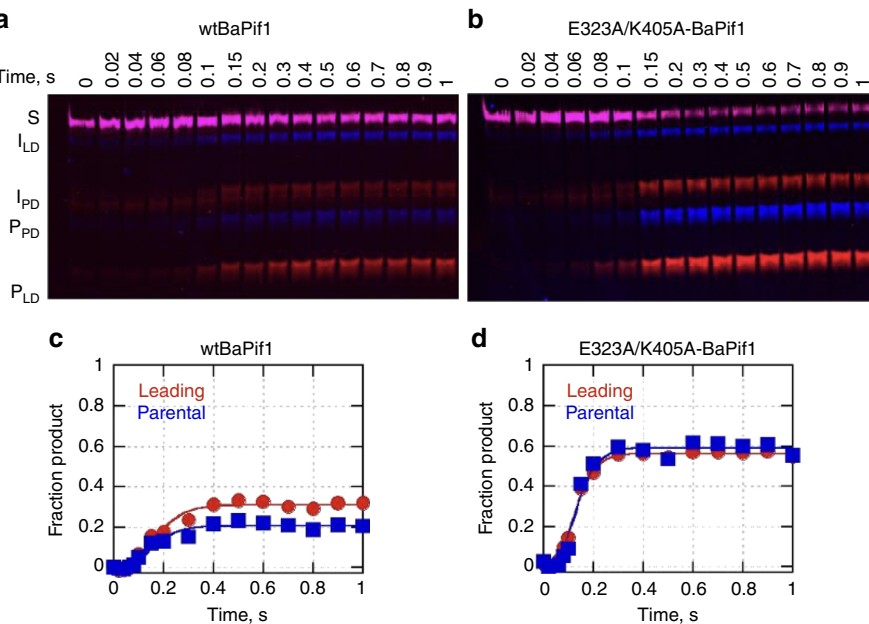

**Fig. 6** Rapid chemical quench flow reactions to measure rate of unwinding the parental and the leading duplexes. **a** Single turnover unwinding assay with wtBaPif1 and E323A/K405A (**b**) to determine rates of unwinding. **c** Quantification of the unwinding reaction with wild type BaPif1. The leading duplex was unwound (circles) at $21 \pm 2\,s^{-1}$ and the parental duplex (squares) at $21 \pm 3\,s^{-1}$. **d** Quantification of the unwinding reaction with E323A/K405A mutant of BaPif1: The mutant unwound the leading duplex (circles) at $33 \pm 1\,s^{-1}$ and the parental duplex (squares) at $31 \pm 3\,s^{-1}$. Errors are the standard error of the fit. $n = 2$ independent experiments. Source gels and quantification data are provided as a Source Data file.

showing two interacting SF1 helicases that are bound at forked dsDNA, with one monomer bound to the 3′-arm and the other to the 5′-arm, yet both are active on their respective strands.

## Discussion

In eukaryotes, the progression and integrity of DNA replication fork (RF) are tightly regulated to maintain genomic stability and for coupling DNA synthesis with other processes. Many physical impediments such as protein–DNA complexes, replication fork barriers (RFB) and DNA damage can hinder the DNA replication fork progression[41]. ScPif1 and Rrm3 in *Saccharomyces cerevisiae* have been shown to function in DNA replication[15,42,43]. Most recently, Deegan et al.[44] showed that both ScPif1 and Rrm3 stimulate fork convergence by helping to unwind the final stretch of parental DNA during DNA replication termination[44]. If BaPif1 functions similarly to ScPif1, then two helicase molecules at the fork should help clear both strands for replication. Consistent with the findings that ScPif1 and Rrm3 have overlapping functions at replication forks, the C-terminal helicase domains of ScPif1 and Rrm3 are well conserved. It has been proposed that Rrm3 might move with the replication fork,

enhancing the activity of the replicative helicase to enable it to bypass protein–DNA complexes[45] while ScPif1 and Rrm3 may help to overcome geometrical strain at converging forks to unwind the final stretch of parental dsDNA[44]. Our structure showed that one BaPif1 binds to the 3′ arm, corresponding to the leading strand template of a fork and the other BaPif1 binds to the 5′ arm, corresponding to the lagging strand template of a fork. The BaPif1 bound to the leading strand template, on one hand, would displace the protein–DNA complexes by its translocase activity; on the other hand, it would enhance parent duplex unwinding by assisting the BaPif1 bound to the lagging strand template. It is envisaged that the coordinating actions of two BaPif1 molecules bound to the replication fork would enable efficient fork progression and convergence. Although ScPif1, Rrm3, and hPif1share low sequence similarity with BaPif1, the structural and biochemical studies on ScPif1 and BaPif1 indicated that they are functionally similar. Therefore, the functional insights we gained from BaPif1 in DNA replication are likely applicable to other Pif1 family helicases.

Many reported activities for Pif1 do not require unwinding of long stretches of duplex DNA. In addition to helping initiate

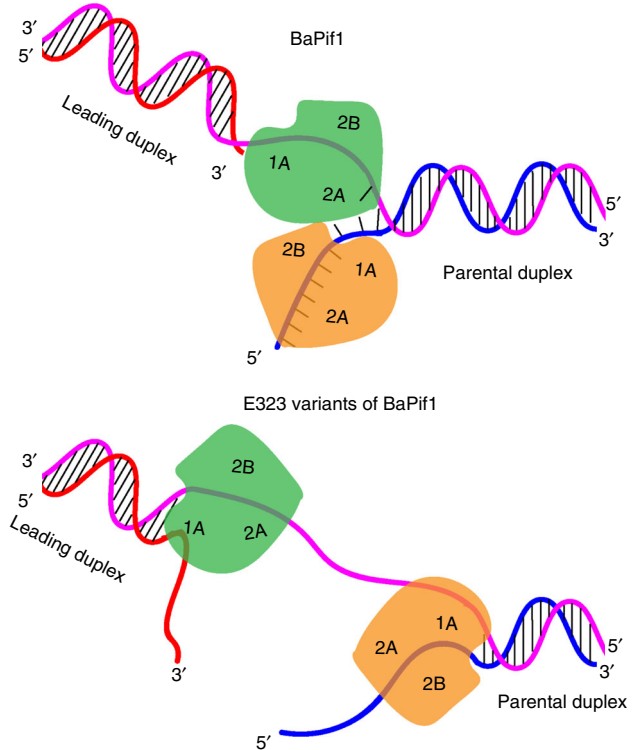

**Fig. 7** Mechanism of unwinding forked dsDNA by BaPif1 helicases. Two molecules of BaPif1 bind to a replication fork; one (orange) binds to the 5′ arm of the fork on the lagging strand template. This BaPif1 bends the ssDNA, which melts the first base pair of the duplex. The other (green) binds to the 3′ arm of the fork on the leading strand template and stabilizes the ss/ds junction. Interactions between the two BaPif1 molecules occur at the junction and appear to limit the processivity of the individual BaPif1 molecules. Upon ATP hydrolysis, each BaPif1 molecule can unwind DNA in a 5′ to 3′ direction. When the interactions in the dimer interface for the E323 variants are reduced, the BaPif1dimer may separate and each molecule may unwind DNA independently.

duplex unwinding, the protein interactions of two Pif1 molecules at a DNA fork could serve a regulatory role by influencing processivity. The removal of secondary structures and bound proteins by Pif1 could be limited to the local region of the replication fork. Pif1 dependent removal of telomerase[6–8] and generation of long flaps during Okazaki fragment maturation[11,12] do not require unwinding long stretches of dsDNA. Disruption of the protein-protein interactions observed here between Pif1 molecules (as seen in Fig. 5) increases processivity, which is consistent with the idea that each monomer regulates the activity of the other.

In conclusion, our structure of BaPif1 bound to a partially unwound forked dsDNA shows that two BaPif1 molecules bind to and unwind the forked dsDNA. The structure revealed several important features that distinguish BaPif1 from any other known SF1 and SF2 helicases. First, the BaPif1 molecule bound to the 5′ arm initiates the forked DNA unwinding by breaking the first base-pair through geometrical strain while the BaPif1 molecule bound to the 3′ arm plays an accessory role in unwinding through stabilizing the first broken base-pair and engaging the second base-pair in a pre-breaking state. Second, the BaPif1 molecule bound to the 3′ arm is positioned to prevent re-winding of the displaced strand with the tracking strand. Third, the two BaPif1 molecules contact each other through electrostatic interactions, and the nature of such interactions may allow two BaPif1 molecules regulate their unwinding activities. These observations together with kinetics and smFRET data enabled us to propose a

mechanism by which BaPif1 unwinds a forked dsDNA. These results not only explain the previous biochemical observations but also provided a framework for further elucidating the mechanism by which Pif1 helicases unwind DNA/RNA hybrids and G4 DNAs.

## Methods

**Protein purification and crystallization.** Pif1 from *Bacterioides fragilis* (BaPif1) and the mutants of BaPif1: D209A, H236A, F379A, H377A, E323K, Y91A/Y412A, E323A/K405A, and D209A/K405A were prepared, expressed and purified using the protocol described in Zhou et al.[25]. A self-complementary oligonucleotide (sequence: 5′-TTTTTTTTTTTCGCGCGCGCGTTTTTTTTTTT-3′) was annealed to generate a symmetrical dual forked dsDNA (referred to as dT10-fdsDNA). Purified wild type BaPif1 was incubated with the dT10-fdsDNA at a molar ratio of 4:1 in the presence of ATP transition-state analogue ADP·AlF₄⁻. The resulting BaPif1–dT10-fdsDNA complex was purified to homogeneity using size exclusion chromatography and then concentrated to approximately 12 mg ml⁻¹. Crystallization screening for the BaPif1–dT10-fdsDNA was performed at 16 °C using the sitting drop vapor diffusion method. Crystals of BaPif1–dT10-fdsDNA were obtained in a condition containing 80 mM NaCl, 40 mM sodium cacodylate pH 6.0, 35% MPD, and 12 mM spermine.

**Structure determination of BaPif1–dT10-fdsDNA.** X-ray diffraction data were collected at Shanghai Synchrotron Radiation Facility. The crystal structure of BaPif1–dT10-fdsDNA was determined with successive rounds of molecular replacement with PHASER[46] using 1A–2A and 2B domains of BaPif1 (PDB id: 2FHD). The final model was built to acceptable stereo-chemical values with iterative model building and refinement cycles using CCP4[47] and Phenix[48]. The difference (Fo−Fc) map was utilized to identify and build DNA, and ADP·AlF₄⁻ and Mg²⁺ were modeled based on PDB id: 5FHD. Data collection and refinement statistics are given in Table 1. A representative portion of electron density map is shown in Supplementary Fig. 10.

**Oligonucleotides for assays.** Oligonucleotides listed in Supplementary Table 1 were purchased from Integrated DNA Technologies and gel purified by denaturing 20% PAGE, visualized by UV shadowing to excise the band, and electroeluted from the acrylamide using an Elutrap (Whatman). Oligonucleotides were desalted by loading onto on a C₁₈ SepPak (Waters) which had been pre-equilibrated by washing with acetonitrile, H₂O, and 10 mM NH₄OAc. SepPaks were washed with H₂O, and oligonucleotides were eluted with 60% methanol and dried in a Speed

**Table 1 Data collection and refinement statistics (molecular replacement).**

|  | BaPif1–10dT-fdsDNA |
|---|---|
| *Data collection* | |
| Space group | P3₁ 1 2 |
| Cell dimensions | |
| *a, b, c* (Å) | 131.3 131.3, 117.2 |
| α, β, γ (°) | 90, 90, 120 |
| Resolution (Å) | 43.7–3.3 (3.56–3.30) |
| $R_{merge}$ | 0.16 (1.0) |
| $I/\sigma I$ | 6.3 (2.2) |
| Completeness (%) | 97.4 (99.6) |
| Redundancy | 4.1 (4.1) |
| *Refinement* | |
| Resolution (Å) | 33.5–3.3 (3.50–3.30) |
| No. reflections | 17075 (2753) |
| $R_{work}/R_{free}$ | 0.24/0.30 (0.33/0.37) |
| No. atoms | |
| Protein and DNA | 7039 |
| Ligand/ion | 66 |
| Water | 5 |
| *B-factors* | |
| Protein and DNA | 83.7 |
| Ligand/ion | 74.9 |
| Water | 57.6 |
| R.m.s. deviations | |
| Bond lengths (Å) | 0.003 |
| Bond angles (°) | 0.6 |

Values in parentheses are for highest-resolution shell

Vac. After re-suspending samples in 10 mM Hepes pH 7.5, 1 mM EDTA, the concentrations were determined using extinction coefficients at 260 nm. Cy5-labeled oligonucleotides which were purchased HPLC and were not gel purified. Duplexes were formed by mixing 5 µM of each strand in 10 mM Hepes pH 7.5, 1 mM EDTA, heating to 95 °C for 5 minutes, and slowly cooling to room temperature.

**DNA unwinding**. BaPif1 (2 µM) was pre-incubated with 20 nM fluorescently labeled dual duplex DNA in 10 mM Tris, pH 7.5, 50 mM NaCl, 0.1 mM EDTA, 2 mM DTT, 0.1 mg ml$^{-1}$ BSA, 5% glycerol for 5 min. Reactions were initiated by addition of ATP to 5 mM, MgCl$_2$ to 10 mM, and 400 nM of annealing trap complementary to the displaced strands at 25 °C. For single turnover reactions, dT$_{50}$ (100 µM) was added with the ATP to prevent re-association of the enzyme with the substrate after dissociation. Reactions were quenched by mixing with an equal quantity of 100 mM EDTA, 0.3% SDS, 1 µM dT$_{50}$, 5% glycerol, 0.05% orange G and separated by 20% native PAGE. Some reactions were performed using a rapid chemical quench flow instrument. For these experiments, reactions were quenched with 400 mM EDTA and loading buffer was added to 5% glycerol, 0.05% orange G. Gels were visualized with a Typhoon Trio using a 488 nm laser for excitation and a 520 BP 40 filter for emission of FAM and a 633 nm laser for excitation and a 670 BP 30 filter for emission of Cy5. The blue and red channels of the gels were quantified separately using ImageQuant Software and the fraction product for each duplex was determined using Equation 1.

$$\text{Fraction product} = \frac{\frac{P_t}{S_t + I_t + P_t} - \frac{P_0}{S_0 + I_0 + P_0}}{1 - \frac{P_0}{S_0 + I_0 + P_0}} \qquad (1)$$

$P_t$ is the fraction product at a time point, $P_0$ is the fraction product at the initiation of the reaction. $S_t$ is the fraction substrate at a time point, $S_0$ is the fraction substrate at the initiation of the reaction. $I_t$ is the fraction intermediate with the other duplex unwound at a time point, $I_0$ is the fraction intermediate with the other duplex unwound at the initiation of the reaction. Data were fit to a 5-step sequential mechanism using KinTek Explorer software.

**smFRET experiments**. Oligonucleotides
(5′-/5Biotin/ CCAGGCGACATCAGCGTTTTTTTGCAGTGACCAGACAGG-3′; 5′-CCTGTCTGGTCACTGC/3Cy3/-3′; and 5′-/5Cy5/TTTTTTTTTTTCGCTGATGTCGCCTGG-3′) were purchased from Integrated DNA Technologies and annealed by mixing 10 µM of the biotin strand and 11 µM of the Cy3 and Cy5 strands in 10 mM Tris-HCl pH 7.5, 50 mM NaCl, and 150 mM KCl. The mixtures were incubated at 95 °C for 2 min then slowly cooled to 37 °C at a rate of 2 °C per minute, and then cooled to room temperature at a rate of 5 °C per minute. Single-molecule FRET flow experiments were performed on a prism-type total internal reflection fluorescence (TIRF) microscope at room temperature (23 ± 1 °C). The microscope, flow chamber preparation, and DNA substrate immobilization were done as described[49]. The unwinding reaction buffer consisted of 20 mM Tris-HCl pH 7.5, 50 mM NaCl, 5 mM MgCl$_2$, 2 mM DTT, and 2% glycerol. For photobleaching and blinking reduction during imaging, buffers also contained 0.8% (w/v) dextrose, 165 U/mL glucose oxidase, 2170 U/mL catalase, and 2–3 mM Trolox. 100 nM Pif1 and 250 µM ATP, or 25 nM Pif1 and 6.25 µM ATP, were flowed into the sample chamber simultaneously and imaged at an exposure time of 200 ms. 532 nm and 633 nm laser excitations were alternating for every frame during the course of a 100 s movie. Single-molecule trajectories were generated by IDL particle tracker and analyzed by MATLAB (R2015b) scripts to determine the unwinding time for parent and leading duplexes.

**Reporting summary**. Further information on research design is available in the Nature Research Reporting Summary linked to this article.

## Data availability

The atomic coordinates and structural factors for the BaPif1–10dT-fdsDNA complex have been deposited with the Protein Data Bank under accession code 6L3G. The source data underlying Figs. 4c–e, 6a–d and Supplementary Figs. 3b–d, 4a, b, 7b, c, 9b, c, f, h, i are provided as a Source Data file. All data supporting the findings of this study are available from the corresponding authors on reasonable request.

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

## Acknowledgements

We would like to thank the beamline scientists at BL17U of Shanghai Synchrotron Radiation Facility in China for assistance and access to synchrotron radiation facilities. This work was supported by Natural Science Foundation of China (Grant No: 31670820) and the Agency for Science, Technology and Research in Singapore and the National Institutes of Health (R35GM122601).

## Author contributions

H.S., K.D.R., and T.H. conceived and coordinated the study. N.S., A.K.B., S.R.B., O.Y., Y.J., and X.T. performed the experiments. N.S., A.K.B., S.R.B., O.Y., T.H., K.D.R., and H.S. analyzed the data and wrote the paper. S.R.B. and O.Y. contributed equally to this work.

## Competing interests

The authors declare no competing interests.
