## [Peer Review File · Nature Communications]

Reviewers' comments:

Reviewer #1 (Remarks to the Author):

Pif1 family helicases are conserved throughout prokaryotes and eukaryotes where they play important roles in genome maintenance. Crystal structures of Pif1 helicases have been determined previously in the presence and absence of ssDNA. The principle advance in the manuscript by Su, et al ("Structural Basis for DNA Unwinding at Forked dsDNA by two coordinating Pif1 helicases") is the determination of a Pif1 structure bound to a forked dsDNA substrate – a substrate that more closely resembles that encountered by Pif1 in vivo.

The structure shows two Pif1 molecules bound to the ss/dsDNA junction in an asymmetrical manner, with one molecule interacting primarily on the 3' strand and the other on the 5' strand. Analysis of unwinding rates suggests that both Pif1 molecules are simultaneously active. Significantly, mutational analysis of the protein-protein interface results in increased unwinding activity, indicating that the activity is modulated by the interaction of the two Pif1 molecules.

I have no criticisms of the data or the presentation of the data. The crystal structure appears to be well refined, given the modest resolution of the data, and the data are appropriately presented. Similarly, the unwinding analysis and FRET experiments appear to be performed properly (although the FRET analysis is somewhat beyond my specific expertise). The manuscript is well written and easily accessible. The work will be of broad interest to the community.

The authors reach a reasonable conclusion based on mutational analysis that the interaction between the two Pif1 molecules is responsible for the observed regulation of unwinding activity. A specific role for E323 is identified. However, the authors make no attempt to explain how modulation occurs. Can the authors propose a model for how the protein-protein interaction modulates activity, including a role for E323?

Minor comment: pg 7, line 153 – a space is missing in "Van der Waal's"

Reviewer #2 (Remarks to the Author):

In this work Su et al present the crystal structure of a dimer of BaPif1 bound to a fork DNA substrate. This is an important contribution to the field, providing a first example of a dimer of a helicase bound to an unwinding substrate. Furthermore, the authors provide ensemble and single molecule unwinding data, for both wild-type and selected mutants, from which they build a model of how a dimer of BaPif1 may work. The data for BaPif1 suggest that this helicase preferentially unwinds the leading strand compared to the parental one, with each monomer bound in the 5'-3' orientation on each ssDNA at the fork (as shown in the structure) and, possibly, working separately of the two duplex regions. The manuscript is clearly written, and the data provide overall support to the authors' conclusions. Some questions and comments are as follow:

1) In the section "Mechanism of forked dsDNA unwinding by BaPif1" the authors omit the wording Ba in front of Pif1, thereby giving the impression that the model applies to Pif1 helicases in general. However, from the data in Figure S8 it is not clear whether this same mechanism applies to human or yeast Pif1. For the latter, the cited studies (lines 312-314) suggest that formation of a dimer on DNA is stimulatory for unwinding activity, a finding that one may not expect if the monomer on the 3' ssDNA region of the DNA were to move away from the fork. Furthermore, the interactions between the A and B monomers are not well conserved (line 55, Fig. S2).

2) In their description of the model the authors write "two BaPif1 molecules bound to the fork, initiate unwinding, after which each of two BaPif1 monomers can function separately as an active

helicase" (line 311, 312). This statement leaves the reader with the strong impression, further corroborated by the cartoon in Figure 7, that the dimer breaks. The mutation E323A suggests that changes at the monomer-monomer interface increase unwinding activity. However, no corroborating evidence that dimerization on DNA is weakened is provided.

3) For mutations H236A and F379A the authors show both ATPase and unwinding data. However, this is not the case for the other mutations studied. This becomes important when examining their effect on the unwinding activity. For example, for H236A and F379A the authors write that the reduced ATPase activity may explain the low unwinding. The same may be true for the Y91A/Y412A mutant; however, the authors suggest that the reduced unwinding activity may be due to their possible effect on the DNA conformation at the ss/dsDNA junction. Please provide ATPase data for this double mutant. Similarly, can the increase of unwinding activity of E323A/K405A (or E323K) be explained by an effect on the ATPase activity?

4) The comparison of F379 of BaPif1 with Phe and Tyr in PcrA and UvrD is confusing. In PcrA and UvrD the monomers are oriented towards the dsDNA to be unwound, for F379 the monomer of BaPif1 on the 3' arm would be in the opposite orientation. Where is the correlation, aside from the fact that all these interactions are stacking interactions?

Reviewer #3 (Remarks to the Author):

Pif1 is an import helicase with multiple roles in genome maintenance. In this work, the authors report the first structure of Pif1 in complex with forked dsDNA, elucidating the unwinding mechanisms of the protein. The insights obtained from structural studies are well supported by kinetics and single-molecule FRET studies. I think that this work is a significant advance over what exists already in the literature, recommend the publication of the paper on Nature Communications.

Reviewer #1 (Remarks to the Author):

Pif1 family helicases are conserved throughout prokaryotes and eukaryotes where they play important roles in genome maintenance. Crystal structures of Pif1 helicases have been determined previously in the presence and absence of ssDNA. The principle advance in the manuscript by Su, et al (“Structural Basis for DNA Unwinding at Forked dsDNA by two coordinating Pif1 helicases”) is the determination of a Pif1 structure bound to a forked dsDNA substrate – a substrate that more closely resembles that encountered by Pif1 in vivo.

The structure shows two Pif1 molecules bound to the ss/dsDNA junction in an asymmetrical manner, with one molecule interacting primarily on the 3' strand and the other on the 5' strand. Analysis of unwinding rates suggests that both Pif1 molecules are simultaneously active. Significantly, mutational analysis of the protein-protein interface results in increased unwinding activity, indicating that the activity is modulated by the interaction of the two Pif1 molecules.

I have no criticisms of the data or the presentation of the data. The crystal structure appears to be well refined, given the modest resolution of the data, and the data are appropriately presented. Similarly, the unwinding analysis and FRET experiments appear to be performed properly (although the FRET analysis is somewhat beyond my specific expertise). The manuscript is well written and easily accessible. The work will be of broad interest to the community.

The authors reach a reasonable conclusion based on mutational analysis that the interaction between the two Pif1 molecules is responsible for the observed regulation of unwinding activity. A specific role for E323 is identified. However, the authors make no attempt to explain how modulation occurs. Can the authors propose a model for how the protein-protein interaction modulates activity, including a role for E323?

We thank the reviewer for in-depth and positive comments.

Mutations of E323 reduces interactions at the protein-protein interface, thereby increasing the ability of the monomers to translocate unhindered in the forward direction. This results in more processive unwinding by the two E323 variants. We have revised the model and the description in the text to indicate this.

Minor comment: pg 7, line 153 – a space is missing in “Van der Waal’s”

We have made the correction accordingly.

Reviewer #2 (Remarks to the Author):

In this work Su et al present the crystal structure of a dimer of BaPif1 bound to a fork DNA substrate. This is an important contribution to the field, providing a first example of a dimer of a helicase bound to an unwinding substrate. Furthermore, the authors provide ensemble and single molecule unwinding data, for both wild-type and selected mutants, from which they build a model of how a dimer of BaPif1 may work. The data for BaPif1 suggest that this helicase preferentially unwinds the leading strand compared to the parental one, with each monomer bound in the 5'-3' orientation on each ssDNA at the fork (as shown in the structure) and, possibly, working separately of the two duplex regions. The manuscript is

clearly written, and the data provide overall support to the authors' conclusions. Some questions and comments are as follow:

We thank the reviewer for favorable and insightful comments.

1) In the section "Mechanism of forked dsDNA unwinding by BaPif1" the authors omit the wording Ba in front of Pif1, thereby giving the impression that the model applies to Pif1 helicases in general. However, from the data in Figure S8 it is not clear whether this same mechanism applies to human or yeast Pif1. For the latter, the cited studies (lines 312-314) suggest that formation of a dimer on DNA is stimulatory for unwinding activity, a finding that one may not expect if the monomer on the 3' ssDNA region of the DNA were to move away from the fork. Furthermore, the interactions between the A and B monomers are not well conserved (line 55, Fig. S2).

In order to clarify the species of Pif1, we have prefixed Ba or Sc or h with Pif1 as appropriate throughout the manuscript.

2) In their description of the model the authors write "two BaPif1 molecules bound to the fork, initiate unwinding, after which each of two BaPif1 monomers can function separately as an active helicase" (line 311, 312). This statement leaves the reader with the strong impression, further corroborated by the cartoon in Figure 7, that the dimer breaks. The mutation E323A suggests that changes at the monomer-monomer interface increase unwinding activity. However, no corroborating evidence that dimerization on DNA is weakened is provided.

We agree with the reviewers' comment regarding the implications of our statement and model. In response, we have re-written the relevant text and revised the model. Indeed, we do not have evidence that the wtPif1 dimer breaks apart (as was shown in our original model). The original model, with the dimer breaking apart, may apply for the E323A mutant due to reduced protein-protein interactions. In our new model, we maintain the dimer interface with wtPif1, and show the dimer breaking with E323 variant. In support of our new model, gel filtration analysis of wtBaPif1 or E323K variant mixed with a single forked DNA at a molar ratio of 2:1 (protein:DNA) in the presence of ADP·AlF₄⁻ showed that less 2:1 and more 1:1 BaPif1-DNA complexes were formed for E323K than WT BaPif1 (Supplementary Figure 7), indicating that dimerization on DNA is weakened for the E323K variant.

New text in the "Mechanism of forked dsDNA unwinding by BaPif1" section reads as follows: "Two BaPif1 molecules bound to the fork initiate unwinding, after which each BaPif1 monomer can function as an active helicase. However, the protein dimer ensures only short stretches of duplex are unwound, keeping with many of the reported functions of Pif1 family enzymes."

3) For mutations H236A and F379A the authors show both ATPase and unwinding data. However, this is not the case for the other mutations studied. This becomes important when examining their effect on the unwinding activity. For example, for H236A and F379A the authors write that the reduced ATPase activity may explain the low unwinding. The same may be true for the Y91A/Y412A mutant; however, the authors suggest that the reduced unwinding activity may be due to their possible effect on the DNA conformation at the ss/dsDNA junction. Please provide ATPase data for this double mutant. Similarly,

can the increase of unwinding activity of E323A/K405A (or E323K) be explained by an effect on the ATPase activity?

We have added ATPase data for Y91A/Y412A, E323A/K405A, and E323K (Supplementary Figure 4b).

BaPif1 is an ATP-dependent helicase and ATP hydrolysis-dependent translocation drives the 5'-3' unwinding. The ATPase rate of Y91A/Y412A was reduced relative to the wild type enzyme. This result is not surprising given that Y91 and Y412 stabilize the bent conformation of ssDNA at 5' ss/dsDNA junction (Figure 2) As the ssDNA bending caused by BaPif1 bound to the 5' arm is a key event, mutation of these two Tyr residues to Ala would be expected to destabilize the bent conformation of substrate DNA, thereby impacting both the ATPase and unwinding activities of BaPif1.

We have also measured the ATPase activity of the E323 variants and found a slight increase in activity compared to the wild type enzyme (Supplementary Figure 4b). We have added discussion of these results to the text in the following sections: "Mutational analysis of residues interacting with ss/dsDNA junction" and "Mutational analysis of residues in the BaPif1-BaPif1 interface".

4) The comparison of F379 of BaPif1 with Phe and Tyr in PcrA and UvrD is confusing. In PcrA and UvrD the monomers are oriented towards the dsDNA to be unwound, for F379 the monomer of BaPif1 on the 3' arm would be in the opposite orientation. Where is the correlation, aside from the fact that all these interactions are stacking interactions?

We have added additional description to this section of the manuscript that we hope will clarify our point.

The similarity of the interactions of BaPif1^B on the 3'-arm of the fork with the interactions of PcrA and UvrD with a fork was initially surprising to us since PcrA and UvrD would translocate toward the fork while BaPif1^B would translocate away from the fork. The conservation of this interaction among helicases of different families with different directionalities is intriguing, and we have tried to clarify this in the text in the "Mutational analysis of residues interacting with ss/dsDNA junction" section.

Reviewer #3 (Remarks to the Author):

Pif1 is an import helicase with multiple roles in genome maintenance. In this work, the authors report the first structure of Pif1 in complex with forked dsDNA, elucidating the unwinding mechanisms of the protein. The insights obtained from structural studies are well supported by kinetics and single-molecule FRET studies. I think that this work is a significant advance over what exists already in the literature, recommend the publication of the paper on Nature Communications.

Thank you for your positive reviews of our manuscript.

REVIEWERS' COMMENTS:

Reviewer #1 (Remarks to the Author):

The authors have satisfactorily addressed all of the comments in my review.

Reviewer #2 (Remarks to the Author):

Inclusion of additional data in Figure S4b and S7 and the changes to the text clarify the concerns raised in the first review and further strengthen the conclusions of this work. No further change/modification is requested.